# Hybrid Modeling of Machine Learning and Phenomenological Model for Predicting the Biomass Gasification Process in Supercritical Water for Hydrogen Production

**Julles Mitoura dos Santos Junior \*** , **Ícaro Augusto Maccari Zelioli \*** and **Adriano Pinto Mariano \***

School of Chemical Engineering, University of Campinas, Av. Albert Einstein 500, Campinas 13083-852, Brazil
* Correspondence: jullesmitoura7@gmail.com (J.M.d.S.J.); i.augustomz@gmail.com (Í.A.M.Z.);
  adpm@unicamp.br (A.P.M.)

**Abstract:** Process monitoring and forecasting are essential to ensure the efficiency of industrial processes. Although it is possible to model processes using phenomenological approaches, these are not always easy to apply and generalize due to the complexity of the processes and the high number of unknown parameters. This work aims to present a hybrid modeling architecture that combines a phenomenological model with machine learning models. The proposal is to enable the use of simplified phenomenological models to explain the basic principles behind a phenomenon. Next, the data-oriented model corrects deviations from the simplified model predictions. The research hypothesis consists of showing the benefits of integrating prior knowledge of chemical engineering in simplifying data-based models, enhancing their generalization and improving their interpretability. The gasification process of lignin biomass with supercritical water was used as a case study for this methodology and the variable to be observed was the production of hydrogen. The real experimental data of this process were augmented using Gibbs energy minimization with the Peng–Robinson equation of state, thus generating a more voluminous database that was considered as real process data. The ideal gas model was used as a simplified model, producing significant deviations in predictions (relative deviations greater than 20%). Deviations ($\Delta H_2 = H_2^{real} - H_2^{predict}$) were used as the target variable for the machine learning model. Linear regression models (LASSO and simple linear regression) were used to predict $\Delta H_2$ and this variable was added to the simplified forecast model. This consisted of the hybrid prediction of the resulting hydrogen formation ($H_2^{predict}$). Among the verified models, the simple linear regression adjusted better to the values of $\Delta H_2$ ($R^2 = 0.985$) and MAE smaller than 0.1. Thus, the proposed hybrid architecture allowed for the prediction of the formation of hydrogen during the gasification process of lignin biomass, despite the thermodynamic limitations of the ideal gas model. Hybridization proved to be robust as a process monitoring tool, providing the abstraction of non-idealities of industrial processes through simple, data-oriented models, without losing predictive power. The objective of the work was fulfilled, presenting a new possibility for the monitoring of real industrial processes.

**Keywords:** SCWG; hydrogen; machine learning; phenomenological model; hybrid model

## 1. Introduction

Modern engineering seeks the optimized use of raw materials and resources, and a common way to achieve such goals is through rigorous process monitoring. Chemical processes, in general, play a significant role in control and monitoring systems to ensure the proper development, to avoid waste, and to maximize the efficiency of processes [1].

For processes with chemical reactions, control can be hampered in most cases because of their nonlinear nature. The reason may be related to the complexity of the reaction system, where numerous intermediate products can be formed throughout the reaction process. Biomass gasification processes in supercritical water are examples of reaction

systems where the reaction's behavior prediction may be hampered due to the complexity of the reaction system [2].

The biomass gasification process in supercritical water reaches good levels of hydrogen formation [3,4]. However, it consists of a complex reaction system, which justifies the need for good monitoring of the operational variables of this process [5].

*The Process of Gasification of Biomass in Supercritical Water*

The energy matrix of the current socioeconomic model is heavily dependent on fossil sources, and conventional use has become extremely expressive since the first industrial revolution. These are non-renewable energy sources, and their use generates significant levels of polluting emissions [6]. In this context, the search for energy sources with lower environmental impact has become one of the main objectives of modern engineering.

A good example of an alternative energy source is hydrogen, which gained attention in the first decade of the 21st century [7]. Hydrogen has a low environmental impact and a high energy density, in addition to having several applications [8]. Because of its high energy density and wide field of applications, there is a constant effort to search for processes to produce hydrogen to compensate for future energy needs and to improve the efficiency of existing processes. Among the routes for obtaining hydrogen, those which convert biomass have been gaining visibility due to their flexibility of application and the availability of biomass sources.

The process of converting biomass into hydrogen using supercritical water as the reaction medium is among the most promising routes. Water acts as a hydrogen donor for the reaction medium, and thus it is possible to gasify biomass with high humidity, which eliminates the need for pre-sucking processes, as is required in conventional processes. This consists of an opportunity to recover the energy of many residues and organic by-products [9].

Despite its relevance, gasification with supercritical water is a complex process, with numerous possible intermediate reactions that may involve the formation of numerous intermediate components in different phases. In addition, the high heterogeneity of biomass hinders the construction of a generalized monitoring model that accurately describes all the details of the mechanism [10,11].

Guan et al. [12] presented a kinetic model to describe the reaction mechanism of the gasification process of microalgae biomass and a supercritical medium. Equations (1)–(12) present the reaction mechanism proposed by them. Biomass consists of a set of macro-molecules, which are quickly decomposed into smaller molecules in a supercritical medium. After, they are converted into gaseous products.

$$\text{Algae} \xrightarrow{k_1} \text{Int.1} \tag{1}$$

$$\text{Algae} \xrightarrow{k_2} \text{Int.2} \tag{2}$$

$$\text{Int.i} + 0.57\ H_2O \xrightarrow{k_{i1}} CO + 1.43\ H_2 \tag{3}$$

$$\text{Int.i} + 1.57\ H_2O \xrightarrow{k_{i2}} CO + 2.43\ H_2 \tag{4}$$

$$\text{Int.i} \xrightarrow{k_{i3}} CO \tag{5}$$

$$\text{Int.i} \xrightarrow{k_{i4}} CO_2 \tag{6}$$

$$\text{Int.i} \xrightarrow{k_{i5}} CH_4 \tag{7}$$

$$\text{Int.i} \overset{k_{i6}}{\rightarrow} H_2 \tag{8}$$

$$\text{Int.i} \overset{k_{i7}}{\rightarrow} C_2 H_a \tag{9}$$

$$\text{Int.i} \overset{k_{i8}}{\rightarrow} \text{Char} \tag{10}$$

$$CO + H_2O \overset{k_3}{\rightarrow} CO_2 + H_2 \tag{11}$$

$$CO + 3H_2 \overset{k_4}{\rightarrow} CH_4 + H_2O \tag{12}$$

Note the formation of intermediate components. The reaction mechanism presented (Equations (1)–(12)) shows a certain inaccuracy with respect to the process steps, since there is not full knowledge of the possible by-products generated during the reactions, which may make the construction of monitoring tools very challenging.

A common approach in modeling literature in such cases of knowledge incompleteness is to build empirical data-driven models. Industries are an abundant source of data, and they must be used to leverage a company's capacity for self-improvement [13].

Because of the growing complexity of industrial processes, the need for more sophisticated modeling techniques has increased proportionally. Machine learning and artificial intelligence techniques are among the top approaches of interest because of their predictive power and wide area of applicability. Ge et al. [13] and Venkatasubramanian [14] present complete reviews of how these techniques have been applied to help solve chemical engineering problems.

Machine learning techniques have been widely applied by chemical process researchers to monitor process parameters [15–17]. Marciej et al. [18] used the Extreme Gradient Boosting (XGBoost) model for data regression in order to predict the carbon straightening capacity in mixtures. Yang et al. [19] constructed a multi-feature fusion convolutional neural network and Light Gradient Boosting Machine (LightGBM) to monitor the safety of oil and gas pipelines. Zhang at al. [20] used Multilayer Perceptron and Random Forest to model the spontaneous combustion tendencies of coal with respect to crossing point temperature. Azarpour et al. [21] proposed a hybrid model combining a first-principle model and artificial neural network, with the aim of predicting the kinetic constant of deactivation of catalysts in a fixed bed. Lei Y. et al. [22] present a hybrid model proposal using four machine learning models (Artificial Neural Networks, Random Forest, XGBoost and LightGBM) for the prediction of hydrogen and methane in raw coke oven gas, presenting coefficients of determination equal to 0.99952 and 0.99964 for the prediction of hydrogen and methane concentrations, respectively, for the best model (LightGBM). Shahbaz et al. [23] constructed an ANN for the prediction of the palm kernel bark steam gasification process using CaO as adsorbent and coal ash as a catalyst. The authors used the backpropagation algorithm to train seven neurons in the hidden layer. The gas composition predicted by the ANN was compared with real data from the pilot scale process, showing high agreement with $R^2 = 0.998$ for almost all cases.

Despite several applications already reported in the literature on the application of data models for the prediction of chemical processes, a disadvantage of data-oriented models is the difficulty of generalizing correlations outside the original range of training data. This is a special issue in process monitoring because they naturally evolve over time due to changing operating conditions.

The current work proposes the creation of a modeling architecture that takes advantage of both approaches: phenomenological and data-driven. Through their union, a hybrid model is built. The work demonstrates an example of the application of this methodology in the modeling of the gasification reaction system with supercritical water.

## 2. Methodology

### 2.1. Phenomenological Modeling of the Process

For the prediction of the biomass gasification process, the thermodynamic approach of minimization of Gibbs energy (minG) will be used. Any system reaches its thermodynamic equilibrium if the total Gibbs free energy has the smallest possible value, so this objective function is widely applied to verify processes in the equilibrium condition [24].

The Gibbs energy minimization approach has greater advantages because it is a direct minimization method that predicts the formation of the system phases and describes the equilibrium compositions adequately, as shown in the works of Rocha and Guirardello [25], Voll et al. [26], and Hantoko et al. [27]. This method has the advantage of considering, in addition to the conservation of masses and equality of fugacity, the minimum Gibbs energy of the system, making it unnecessary to worry about predicting the possible phases that the system may form [28].

For reactive systems with multiple components conditioned at constant pressures and temperatures, the thermodynamic equilibrium condition can be formulated as a Gibbs energy minimization problem, with the Gibbs energy described by Equation (13).

$$minG = \sum_{i=1}^{NC} \sum_{k=1}^{NF} n_i^k \left[ \mu_i^o + RTln\left( \hat{f}_i^k / f_i^o \right) \right] \tag{13}$$

The direct minimization of Equation (13), considering the restrictions of mass balance and stoichiometry, results in a combined chemical and phase equilibrium point. For the system to reach an adequate solution, it is necessary to add two constraints. The first constraint is the non-negativity of the number of moles, Equation (14), of each of the components in each of the phases [28].

$$n_i^k \geq 0 \tag{14}$$

The second restriction is related to the balance of atoms due to the non-stoichiometric formulation, which does not consider the possible reactions that occur throughout the optimization process, but the best arrangement of atoms is represented by Equation (15).

$$\sum_{i=1}^{NC} \sum_{i=1}^{NF} a_{mi}\left(n_i^k\right) = \sum_{j=1}^{NC} a_{mi}\left(n_i^0\right) \tag{15}$$

When the conservation of matter equation is satisfied, the Gibbs free energy expression obtains its minimum value when a multicomponent system reaches chemical equilibrium [29].

Bearing in mind that gasification processes in supercritical media occur under high pressure and temperature conditions, it is estimated that components in the liquid phase will not be formed; even so, both phases will be considered in the modeling process. Equation (13) can be rewritten in terms of chemical potentials and molar amounts of solid, liquid, and vapor phase components, as described in Equation (16).

$$minG = \sum_{i=1}^{NC} \left( n_i^s \mu_i^s + n_i^v \mu_i^v + n_i^l \mu_i^l \right) \tag{16}$$

The standard chemical potential can be calculated from Equations (17) and (18). These results are necessary for estimating the Gibbs energy, as shown in Equation (16).

$$\frac{\partial}{\partial T} \left( \frac{\mu_i^k}{RT} \right)_P = - \frac{\overline{H}_i^g}{RT^2} \tag{17}$$

$$\left(\frac{\partial \overline{H}_i^g}{\partial T}\right)_P = Cp_i^g \tag{18}$$

To facilitate the thermodynamic modeling of the process, the solid phase will be considered ideal (Equation (19)), so it will not be necessary to estimate non-idealities. This consideration seems to be reasonable, considering that throughout the gasification process with supercritical water, high levels of water are inserted in the reaction system, hindering the formation of components in the solid phase [3,4,28].

$$\mu_i^s = \mu_i^0 \tag{19}$$

Contrary to the hypothesis adopted regarding the ideality of the solid phase, the vapor phase cannot be considered ideal since the conditions of the process in question make this consideration impossible. Equation (20) describes the chemical potential of the components in the vapor phase written as a function of the standard chemical potential, temperature, molar composition in the vapor phase, pressure, and coefficient of fugacity of the components considered.

$$\mu_i^v = \mu_i^0 + RT(\ln \hat{\varnothing}_i^v + ln y_i + \ln P) \tag{20}$$

Equation (21) presents the chemical potential of the components in the liquid phase. This is written as a function of the standard chemical potential, temperature, molar composition in the vapor phase, pressure, and fugacity coefficient of the considered components.

$$\mu_i^l = \mu_i^0 + RT\left(\ln \hat{\varnothing}_i^l + ln x_i + \ln P_i^{sat}\right) \tag{21}$$

The chemical potential of the liquid phase components is calculated as a function of the saturation pressure, and the Antoine equation (Equation (22)) will be used to calculate this property.

$$ln P_i^{sat} = a_i - \frac{b_i}{c_i + T} \tag{22}$$

The Peng–Robinson cubic equation of state (EoS) will be applied to estimate the non-idealities of the liquid and vapor phases [30]. The next section presents in more detail the estimation of fugacity coefficients using the Peng–Robinson EoS.

The molar partial enthalpy of each liquid or gaseous I component is calculated as a function of their heat capacities, which are a function of temperature, as shown in Equation (23).

$$Cp_i^v = A_{0,i} + A_{1,i}T + A_{2,i}T^2 + A_{3,i}T^3 + A_{4,i}T^4 \tag{23}$$

For solids, the heat capacity is calculated according to Equation (24).

$$Cp_i^s = A_i + B_iT + C_iT^2 + D_iT^{-2} \tag{24}$$

The parameters for calculating the saturation pressures and formation properties of the considered components are presented in Table 1. The parameters for calculating the heat capacities of the solid and vapor phase components are presented in Tables 2 and 3, respectively. The reference state of a species in the gas phase is given by the pure substance at 1 bar and system temperature. Liquids and solids use the liquid itself or pure solid at 1 bar [31].

**Table 1.** Critical properties, formation, and parameters of the Antoine equation, as reported by Poling et al. [32].

| Components | $T_c$ (K) | $P_c$ (bar) | $V_c$ (m³/kmol) | $\omega$ | $a$ | $b$ | $c$ | $\Delta H_f$ (cal/mol) | $\Delta G_f$ (cal/mol) |
|---|---|---|---|---|---|---|---|---|---|
| $H_2O$ | 647.140 | 220.640 | 0.056 | 0.344 | 18.304 | 3816.440 | −46.130 | $-5.78 \times 10^4$ | $-5.46 \times 10^4$ |
| $H_2$ | 32.980 | 12.930 | 0.064 | −0.217 | 13.633 | 164.900 | 3.190 | 0 | 0 |
| $CH_4$ | 190.560 | 45.990 | 0.099 | 0.011 | 15.224 | 597.840 | −7.160 | $-1.78 \times 10^4$ | $-1.21 \times 10^4$ |
| $CO_2$ | 304.150 | 73.740 | 0.094 | 0.225 | 22.590 | 3103.390 | −0.160 | $-9.41 \times 10^4$ | $-9.43 \times 10^4$ |
| $CO$ | 132.850 | 34.940 | 0.093 | 0.045 | 14.369 | 530.220 | −13.150 | $-2.64 \times 10^4$ | $-3.28 \times 10^4$ |
| $O_2$ | 154.580 | 50.430 | 0.073 | 0.022 | 15.408 | 734.550 | −6.450 | 0 | 0 |
| $N_2$ | 126.200 | 33.980 | 0.090 | 0.037 | 14.954 | 588.720 | −6.600 | 0 | 0 |
| $CH_4O$ | 512.640 | 80.970 | 0.118 | 0.565 | 18.588 | 3626.550 | −34.290 | $-4.80 \times 10^4$ | $-3.88 \times 10^4$ |
| $C_2H_6$ | 305.320 | 48.720 | 0.146 | 0.099 | 15.664 | 1511.420 | −17.160 | $-2.00 \times 10^4$ | $-7.61 \times 10^3$ |
| $C_3H_8$ | 369.830 | 42.480 | 0.200 | 0.152 | 15.726 | 1872.460 | −25.160 | $-2.50 \times 10^4$ | $-5.81 \times 10^3$ |
| $NH_3$ | 405.400 | 113.530 | 0.072 | 0.257 | 16.948 | 2132.500 | −32.981 | $-1.10 \times 10^4$ | $-3.92 \times 10^3$ |
| $C_2H_4$ | 282.340 | 50.410 | 0.131 | 0.087 | 15.534 | 1347.010 | −18.150 | $1.25 \times 10^4$ | $1.64 \times 10^4$ |

**Table 2.** Coefficients for calculating the heat capacity of solid formation, as reported by Smith et al. [33].

| Components | $A$ * | $B$ * | $C$ * |
|---|---|---|---|
| C | 35.190 | $1.53 \times 10^{-3}$ | $-1.72 \times 10^5$ |
| CaO | 121.286 | $8.80 \times 10^{-4}$ | $-2.08 \times 10^5$ |
| $CaCO_3$ | 249.806 | $5.24 \times 10^{-3}$ | $-6.20 \times 10^5$ |
| $Ca(OH)_3$ | 190.692 | $1.08 \times 10^{-2}$ | 0 |
| NaOH | 0.240 | $3.24 \times 10^{-2}$ | $3.87 \times 10^5$ |

* Values already multiplied by the gas constant (R = 1.987 cal/mol.K).

**Table 3.** Coefficients for calculating the heat capacity of the formation of components in the vapor phase, as reported by Poling et al. [32].

| Components | $A_0$ * | $A_1$ * | $A_2$ * | $A_3$ * | $A_4$ * |
|---|---|---|---|---|---|
| $H_2O$ | 87.329 | $-8.32 \times 10^{-3}$ | $2.79 \times 10^{-5}$ | $-3.11 \times 10^{-8}$ | $1.26 \times 10^{-11}$ |
| $H_2$ | 57.285 | $7.31 \times 10^{-3}$ | $-1.53 \times 10^{-5}$ | $1.38 \times 10^{-8}$ | $-4.23 \times 10^{-12}$ |
| $CH_4$ | 90.766 | $-1.78 \times 10^{-2}$ | $7.21 \times 10^{-5}$ | $-6.77 \times 10^{-8}$ | $2.17 \times 10^{-11}$ |
| $CO_2$ | 64.756 | $2.69 \times 10^{-3}$ | $2.98 \times 10^{-5}$ | $-4.72 \times 10^{-8}$ | $2.10 \times 10^{-11}$ |
| $CO$ | 77.731 | $7.78 \times 10^{-3}$ | $2.35 \times 10^{-5}$ | $-2.59 \times 10^{-8}$ | $1.02 \times 10^{-11}$ |
| $O_2$ | 72.128 | $-3.56 \times 10^{-3}$ | $1.31 \times 10^{-5}$ | $-1.19 \times 10^{-8}$ | $3.56 \times 10^{-12}$ |
| $N_2$ | 70.320 | $-5.19 \times 10^{-4}$ | $1.39 \times 10^{-7}$ | $3.12 \times 10^{-9}$ | $-1.97 \times 10^{-12}$ |
| $CH_4O$ | 93.667 | $-1.39 \times 10^{-2}$ | $8.37 \times 10^{-5}$ | $-8.83 \times 10^{-8}$ | $3.05 \times 10^{-11}$ |
| $C_2H_6$ | 83.017 | $-8.80 \times 10^{-3}$ | $1.12 \times 10^{-4}$ | $-1.32 \times 10^{-7}$ | $4.94 \times 10^{-11}$ |
| $C_3H_8$ | 76.440 | $1.02 \times 10^{-2}$ | $1.19 \times 10^{-4}$ | $-1.57 \times 10^{-7}$ | $6.12 \times 10^{-11}$ |
| $NH_3$ | 84.209 | $-8.38 \times 10^{-3}$ | $4.06 \times 10^{-5}$ | $-4.22 \times 10^{-8}$ | $1.51 \times 10^{-11}$ |
| $C_2H_4$ | 83.880 | $-1.75 \times 10^{-2}$ | $1.15 \times 10^{-4}$ | $-1.34 \times 10^{-7}$ | $4.99 \times 10^{-11}$ |

* Values already multiplied by the gas constant (R = 1.987 cal/mol.K).

Estimation of Fugacity Coefficients Using the Cubic Peng–Robinson Equation

For the prediction of the biomass gasification process from the phenomenological point of view, the thermodynamic approach to minimization of Gibbs energy (minG) will be used. Any system reaches its thermodynamic equilibrium if the total Gibbs free energy has the smallest possible value, so this objective function is widely applied to verify processes in the equilibrium condition [24]. This methodology has great advantages as it is a direct minimization method that predicts the formation of the system phases and satisfactorily describes the equilibrium compositions in reaction systems.

The equations of state can be presented as cubic equations, in the form of the compressibility factor Z, generally described by Equation (25).

$$f(Z) = Z^3 - (1 + B - uB)Z^2 + \left(A + wB^2 - uB - uB^2\right)Z - AB - wB^2 - wB^3 \qquad (25)$$

where $A$ and $B$ are dimensionless dependent on temperature, pressure, and phase composition, as shown in Equations (26) and (27). Parameters $u$ and $w$ are 2 and $-1$, respectively, tabled from Peng–Robinson state approval.

$$A = \frac{a_m P}{(RT)^2} \tag{26}$$

$$B = \frac{b_m P}{RT} \tag{27}$$

where $a_m$ and $b_m$ are mixture properties and determined from Equations (28) and (29), respectively.

$$a_m = \sum_{i=1}^{NC} \sum_{j=1}^{NF} y_i y_j \sqrt{a_i a_j}\left(1 - k_{ij}\right) \tag{28}$$

$$b_m = \sum_{i=1}^{NC} y_i b_i \tag{29}$$

The $k_{ij}$ is a binary interaction parameter and $a_i$ e $a_j$ are parameters that depend on a predetermined constant for each equation of state, the critical properties ($P_c$ and $T_c$), gas constant ($R$), and acentric factor ($\omega_i$) of each component $i$ and $j$. In this way, $a_i$ and $a_j$ are represented by Equation (30).

$$a_i = 0.45724 \frac{R^2 T_{c,i}^2}{P_{c,i}} \alpha_i \tag{30}$$

The parameter $\alpha_i$ is given by Equation (31).

$$\alpha_i = \left[1 + \left(0.37464 + 1.54226\omega_i - 0.26992\omega_i^2\right)\left(1 - \sqrt{\frac{T}{T_{c,i}}}\right)\right]^2 \tag{31}$$

The $b_i$ parameter also depends on the critical properties, gas constant, and acentric factor of each component i, as shown in Equation (32).

$$b_i = 0.07780 \frac{RT_{c,i}}{P_{c,i}} \tag{32}$$

With these data, it is possible to calculate the roots of the cubic equation. The fact that there is only a single real root of the compressibility factor ($Z$) reveals that the mixture exists in a single phase, liquid or vapor. If you have the three real roots, the largest of them will represent the vapor phase and the smallest the liquid phase. The root of the intermediate value has no physical meaning as it violates the mechanical stability criterion [34]. Knowing the root of Equation (25) for both phases, Equation (33) will be used to estimate the fugacity coefficients for the vapor and liquid phases.

$$\ln \hat{\varnothing}_i = \frac{B_i}{B}(Z-1) - \ln(Z-B) + \frac{A}{2\sqrt{2}B}\left(\frac{B_i}{B} - 2\frac{\sum_j y_i \sqrt{a_i a_j}}{a_m}\right)\ln\left(\frac{z + \left(1 + \sqrt{2}\right)B)}{z + \left(1 - \sqrt{2}\right)B}\right) \tag{33}$$

### 2.2. Mathematical Formulation and Solution of the Equilibrium Problem

Equation (25) is known as the cubic equation of state. This equation provides an approximation of the actual behavior of the liquid and vapor region for a series of fluids [31]. The resolution of this equation produces one or three real roots, which can be later used to calculate the fugacity coefficients, in the approach known as phi-phi that will be used in this work.

Authors Kamath, Biegler, and Grossmann [34] determined in their work that the first derivative of the cubic equation of state concerning Z must be positive to avoid selection of the root mean value. Furthermore, the second derivative ensures that the liquid and vapor phase roots are determined. The largest root will determine the vapor phase, whereas the second derivative must be greater than or equal to zero, and the smallest root, which determines the liquid phase, must be less than or equal to zero. Equations (34)–(37) represent these constraints for the Peng–Robinson equation.

$$f'(Z_g) = 3Z_g{}^2 - 2(1 - B)Z_g + A - 2B - 3B^2 \geq 0 \tag{34}$$

$$f'(Z_l) = 3Z_l{}^2 - 2(1 - B)Z_l + A - 2B - 3B^2 \geq 0 \tag{35}$$

$$f''(Z_g) = 6Z_g + 2B - 2 \geq -M\sigma^g \tag{36}$$

$$f''(Z_l) = 6Z_l + 2B - 2 \leq M\sigma^l \tag{37}$$

To avoid selecting a root without physical significance, with the disappearance of one of the phases of the system, with only one phase, gaseous or liquid, Kamath, Biegler, and Grossmann [34] added slack variables ($\sigma^v$ e $\sigma^l$), which are used to allow the program to calculate derivatives when they are equal to zero, as in Equations (33) and (34), obtaining Equations (38) and (39), with modifications for the gaseous and liquid phases, respectively. *M* is a positive and large value. In this work, *M* was considered 10, as well as in the work of Dowling et al. [35].

$$f''(Z_g) = 6Z_g + 2B - 2 \geq -M\sigma^g \tag{38}$$

$$f''(Z_l) = 6Z_l + 2B - 2 \leq -M\sigma^l \tag{39}$$

Initially, 12 components will be considered ($H_2$, $H_2O$, $CH_4$, $CO_2$, $CO$, $O_2$, $N_2$, $CH_4O$, $C_2H_6$, $C_3H_8$, $NH_3$, $C_2H_4$) as representative of the main compounds that it is possible to form during the biomass gasification process in supercritical water. The selection of these components was based on results reported in the literature, which indicate that these are the components formed in considerable compositions during the gasification processes of biomass from different biomass sources [3–5,28,36–41].

The formulated NLP problems will be solved with the aid of the GAMS software and the CONOPT 4 solver, considering that this solver has some advantages about the type of approach that will be used. It is suitable for models with very non-linear constraints, is designed for large models, and can be applied to models that do not have differentiable functions [42]. This approach has demonstrated great accuracy and efficiency and has been used with great results by our research group over the last few years for a wide range of systems under conditions of chemical equilibrium and combined phases [3,4,6,25,26].

Figure 1 presents the proposed algorithm for obtaining the equilibrium compositions throughout the reaction using the Gibbs energy minimization methodology associated with the Peng–Robinson cubic equation of state.

### 2.3. Hybrid Architecture Proposed for the Hybrid Modeling of the Problem

Figure 2 describes the proposed hybrid modeling architecture associating simulated data or data obtained through rigorous modeling with data obtained from a simplified phenomenological model (ideal cases or simplifying hypotheses).

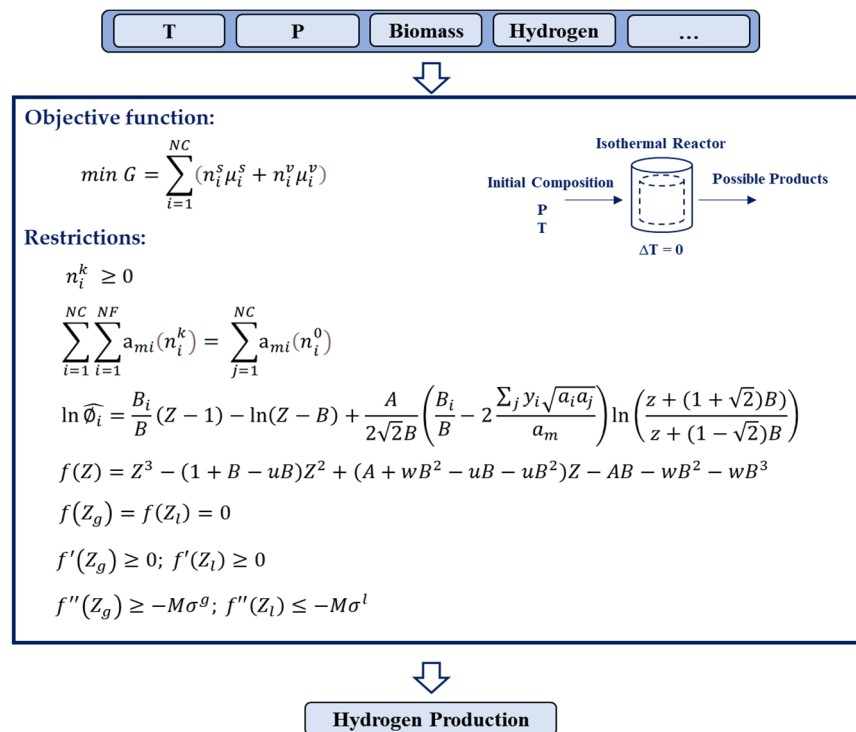

**Figure 1.** Sequential flowchart for predicting the equilibrium compositions of a system using the Gibbs energy minimization methodology with the aid of the cubic Peng–Robinson equation.

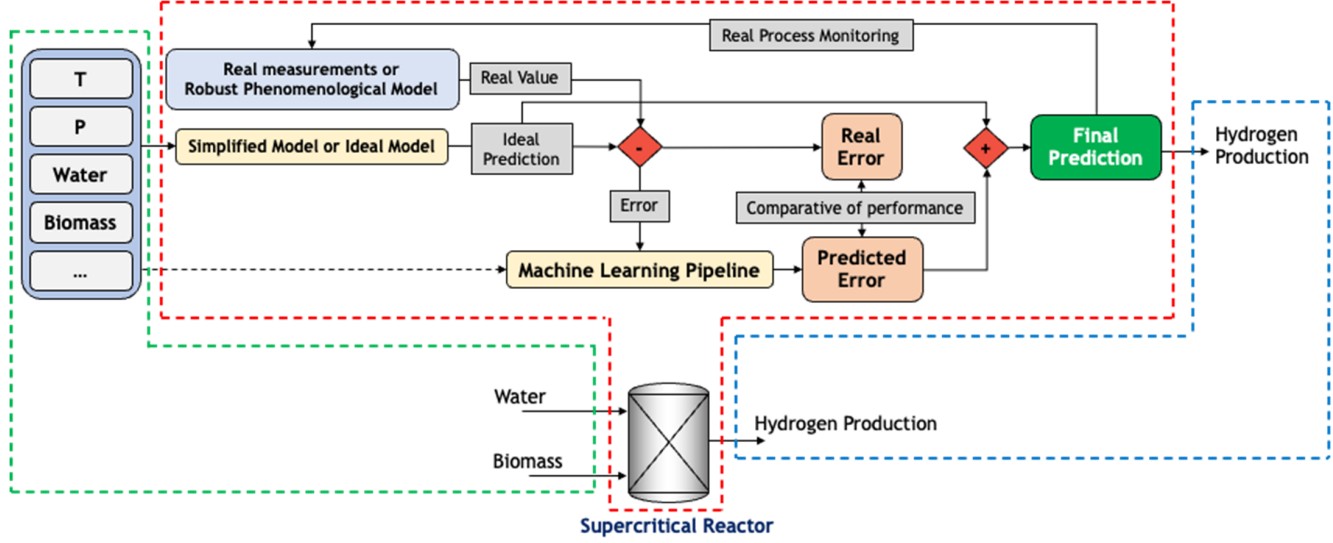

**Figure 2.** Hybrid modeling architecture for predicting the variable of interest in the biomass gasification process with supercritical water.

The hybrid architecture from Figure 2 is based on the concept of boosting as it consists of a set of weak estimators and sequentially organized models that perform a little better than random predictions. Each new estimator is trained to correct the errors made by the previous estimator [43]. The main gain of the proposed approach is to reduce the overall prediction bias.

The first step of the proposed architecture of this work consists in making predictions of the production of hydrogen in the equilibrium condition, considering the system as ideal—i.e., using the Clapeyron equation (Equation (40)). Note that ideal behavior is not

consistent with what is studied, considering that the critical water pressure is greater than 220 bar [6,8,36].

$$PV = nRT \tag{40}$$

The simplified model uses basic inputs to calculate the variable of interest; in this case, the production of hydrogen in the equilibrium condition. Using real process data or data simulated by a more rigorous phenomenological equation, the error of the predictions will be calculated using Equation (41). The second part of the proposed architecture corresponds to the use of a data model that will receive several input values—which may be the same used in an ideal first-principle model—and use them to predict the errors calculated previously.

A set of experimental data reported by Basu [39] will be used for the gasification process of lignin biomass in supercritical water at 30 MPa. Experimental data will be used to validate the methodology described in Section 2.1, using the Gibbs energy minimization methodology associated with the cubic Peng–Robinson equation to calculate non-idealities. Figure 3 presents a comparison of the experimental data reported by Basu [39] with results calculated using the methodology described in Section 2.1.

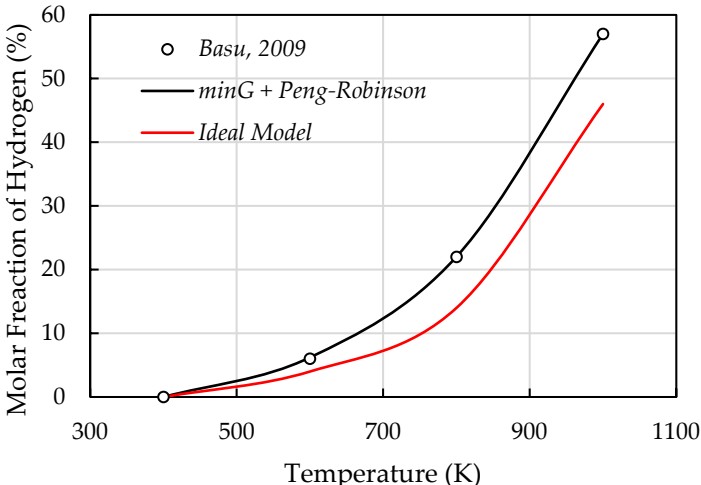

**Figure 3.** Comparison of experimental results reported by Basu (2009) [39] concerning results calculated using the Gibbs energy minimization methodology with the Peng–Robinson equation and the ideal model.

As seen in Figure 3, the thermodynamic modeling applying the minimization of the Gibbs energy associated with the cubic Peng–Robinson equation presents an excellent fit concerning the ideal data, with a mean relative deviation of less than 1.0%. It is also verified that the results obtained considering the ideal model follow the tendency of the molar fraction of hydrogen as a function of temperature; however, the adjustment is not so precise, with an average relative deviation of 22.032%. Hence, from this point on, the results obtained by minimizing the Gibbs energy with the Peng–Robinson equation will be considered as real data.

$$\Delta H_2 = H_2^{real} - H_2^{ideal} \tag{41}$$

Considering that the Gibbs energy minimization methodology with the aid of the Peng–Robinson cubic adjusted well the experimental data of Basu [39], additional data were generated using different conditions of pressure, temperature, and biomass compositions in the feed. Figure 4 represents the described data set expansion procedure.

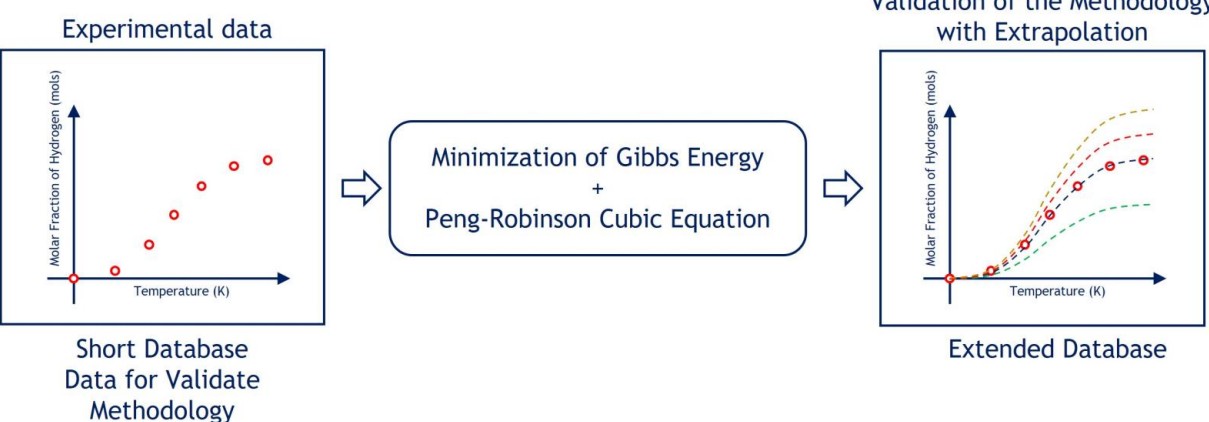

**Figure 4.** The methodology used to expand the data set.

Having the ideal prediction deviation results, the following steps will all be aimed at applying the machine learning model for *KPI* prediction.

The database that will be applied to the machine learning model contains the variables shown in Figure 5.

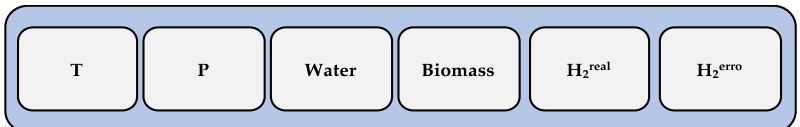

**Figure 5.** Database variables to be used for processing the machine learning model.

The methodology used to expand the data set, as shown in Figure 4, is widely applied to simulations of complex reaction systems. Works reported by Mitoura et al. [6], Gomes et al. [8], and Freitas [28] applied the Gibbs energy minimization methodology to simulate gasification processes of different biomass sources and methane thermal cracking, presenting excellent results.

### 2.3.1. Data Modeling

Considering that one of the objectives of this work is to show the advantages of less complex data approaches, two modeling algorithms were chosen as the first options to model the errors of the ideal model concerning the real data. Two linear regression approaches were selected because they have good generalizability and are easy to interpret [44]. Linear regression models will be used through the LinearRegression class and LASSO regression from the Lasso class, both from the scikit-learn library. Equation (42) presents the generalized form of a linear model.

$$y = B_0 + \sum_{i=1}^{n} B_i x_i \tag{42}$$

where $y$ is the objective variable to be modeled, $B_i$ are the angular coefficients referring to attribute $i$, $B_0$ is the intercept, and $x$ is a predictor variable.

### Attribute Selection, Data Standardization, Model Selection, and Validation

An important procedure in machine learning modeling consists in selecting the attributes that contribute the most to the prediction of a target variable. The main reasons include the existence of multicollinearity effects, which cause redundant information to be inputted in the model. In this work, a simplified approach of feature selection was employed, using only linear correlation as the measure of importance of each feature.

Through the SelectKBest class of Python's scikit-learn library, a linear regression is fitted for each attribute/target pair, and the F statistic is calculated by measuring the goodness of the linear fit. The model selects the attributes that have the highest F statistics [45].

Considering that the attributes have very different scales, another crucial step is to scale the data, which helps to avoid model biases towards features with the widest ranges of variation.

The MinMaxScaler class from the scikit-learn library will be used, which normalizes all features in a single scale (0–1), while keeping their variance. Equation (43) presents the scaling of the data based on their maximum and minimum values.

$$x^{std} = \frac{x - x_{min}}{x_{max} - x_{min}} \tag{43}$$

For the selection of hyperparameters, the RandomizedSearchCV class from the Python scikit-learn package was used, together with the cross-validation strategy using the KFold class from the scikit-learn package. The algorithm was defined to generate 1000 combinations of hyperparameter values. The model selection metric was the mean absolute error (MAE) (Equation (44)), and the coefficient of determination $R^2$ (Equation (45)) was also used as a model selection criterion.

$$MAE = \frac{1}{n} \sum_{j=1}^{n} |\dot{y}_i - \hat{y}_i| \tag{44}$$

$$R^2(\dot{y}, \hat{y}) = 1 - \frac{\sum_{i=1}^{n} (\dot{y}_i - \hat{y}_i)^2}{\sum_{i=1}^{n} (\dot{y}_i - \bar{y}_i)^2} \tag{45}$$

Figure 6 presents the machine learning model pipeline with the descriptions presented.

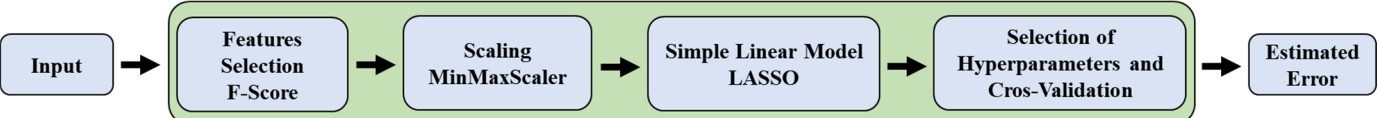

**Figure 6.** Machine learning model pipeline.

The following sections present the results of applying the data-based model for predicting the error between real data and those calculated by the ideal model (Equation (41)). With the estimated error, the corrected hydrogen production will be calculated based on the values predicted by the ideal model, following Equation (46).

$$H_2^{predict} = H_2^{ideal} + \Delta H_2 \tag{46}$$

## 3. Results and Discussions

### 3.1. Presentation of the Database

As mentioned previously, the experimental data from Basu [39] were used to validate the proposed methodology, and after validation, the data set was augmented. Figure 7 shows, as an example, the formation of hydrogen as a function of temperature, fixing 1 mole of biomass with 5 moles of water in the feed for pressures of 300 and 500 bar.

Analyzing Figure 7, the ideal model follows the trend of the real process, even with perceptible deviations. The mean absolute error values are equal to 0.281 and 0.322 for pressures of 300 and 500 bar, respectively. The statistical metrics presented are considerable since the objective of this text is to reduce the bias of a simple first-principle model with the aid of a machine learning model.

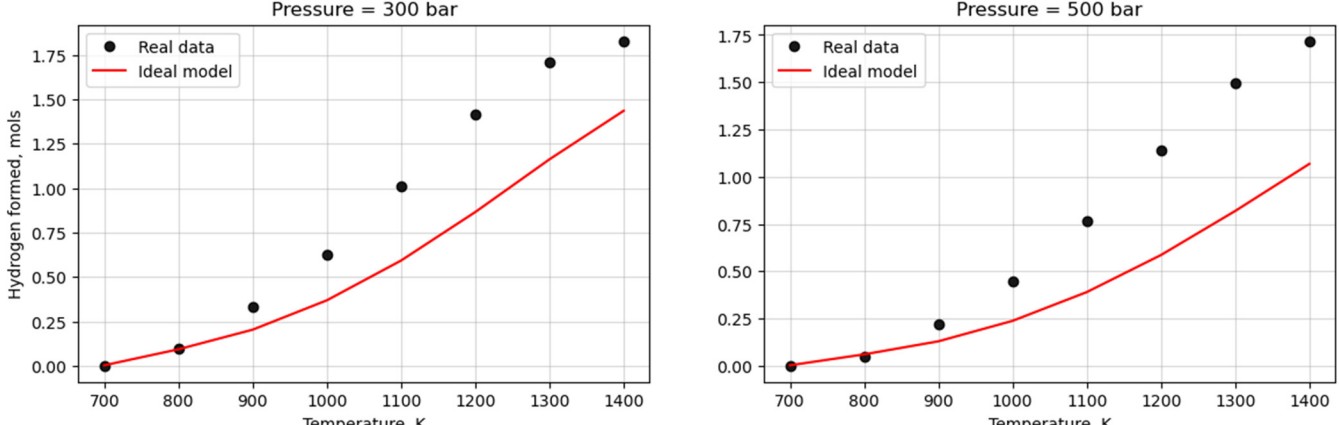

**Figure 7.** Comparison between real and simulated data considering the reaction system as ideal, fixing 1 mole of biomass with 5 moles of water in the feed for pressures of 300 and 500 bar.

To verify the linear correlations between the variables, Figure 8 presents the correlation matrix of the data set. This was an important step because of the types of machine learning models employed (Linear Regression and LASSO).

| | Temperature (K) | Pressure (bar) | Biomass (mols) | Water (mols) | Hydrogen_real (mols) | Hydrigen_ideal (mols) |
|---|---|---|---|---|---|---|
| **Temperature (K)** | 1 | $-1.61 \times 10^{-6}$ | $-6.50 \times 10^{-16}$ | $5.10 \times 10^{-16}$ | 0.86 | 0.91 |
| **Pressure (bar)** | $-1.61 \times 10^{-6}$ | 1 | 0.086 | -0.23 | -0.14 | -0.12 |
| **Biomass (mols)** | $-6.50 \times 10^{-16}$ | 0.086 | 1 | -0.63 | -0.11 | -0.018 |
| **Water (mols)** | $5.10 \times 10^{-16}$ | -0.23 | -0.63 | 1 | 0.27 | 0.17 |
| **Hydrogen_real (mols)** | 0.86 | -0.14 | -0.11 | 0.27 | 1 | 0.98 |
| **Hydrigen_ideal (mols)** | 0.91 | -0.12 | -0.018 | 0.17 | 0.98 | 1 |

| |
|---|
| 1.000 |
| 0.733 |
| 0.467 |
| 0.200 |
| -0.333 |
| -0.600 |

**Figure 8.** Data set correlation matrix.

The temperature has a high positive correlation with the target variable (represented here as "Hydrogen_real"), indicating that the increase in temperature favors the formation of hydrogen throughout the process. This result is expected since it agrees with the kinetic model of Whitag et al. [46], where it is described that in gasification systems in supercritical water, the temperature increase favors the water–gas displacement reactions that form large amounts of hydrogen.

In addition to the effect of temperature, note that the pressure and the biomass feed disfavor the formation of hydrogen. This result is predicted by the model of Whitag et al. [46], where it is described that the increase in pressure disfavors the formation of products of interest throughout the process. This is justified by the fact that the increase in pressure disfavors the water–gas displacement reactions and the methanation reaction is favored, according to Le Chatelier's principle; thus, hydrogen is greatly consumed, forming methane and carbon dioxide.

The models presented by Whitag et al. [46] and Yan et al. [40] describe that the increase in the composition of biomass in the feed harms the formation of hydrogen, while the amount of methane increases. This behavior is justified by the fact that the increase in biomass concentration disfavors the water–gas reactions, which produce greater amounts of hydrogen, which, in turn, favors the methanation reaction, forming methane. The formation of carbon monoxide in low amounts helps to confirm the hypothesis.

Since the increase in biomass composition minimizes the formation of hydrogen, it is expected that the increase in the amount of water in the feed favors the formation of hydrogen, a result that is verified in Figure 8. Water additions to the reaction process favor the reactions of water–gas, increasing the formation of hydrogen, as previously mentioned.

All the above conclusions follow what was predicted by the models presented by Guan et al. [12], Yan et al. [40], Castello and Fiori [47], Goodwin and Rorrer [48], and Tang and Kitagawa [38] for the behavior of biomass gasification processes in supercritical water. In addition to the listed models, recent work reported by Chen et al. [49] and Gomes et al. [8] studying gasification processes of biomass sources using supercritical water as a reaction medium presented results in agreement with those presented in this text.

With the data describing the actual hydrogen production and calculated by the ideal model as a function of the other variables (temperature, pressure, and composition of biomass/water in the feed), the ideal model's deviations presented in Equation (41) were calculated and their correlation matrix was built, as Figure 9 shows. The produced quantity of ideal hydrogen (Hydrogen_ideal) has a high correlation with the temperature, thus the temperature and the molar quantity of ideal hydrogen are collinear. Multicollinearity is a problem in the model's fitting because it can impact the estimation of the parameters [50]. Given the multicollinearity problem, the Hydrogen_ideal variable was removed from the data set.

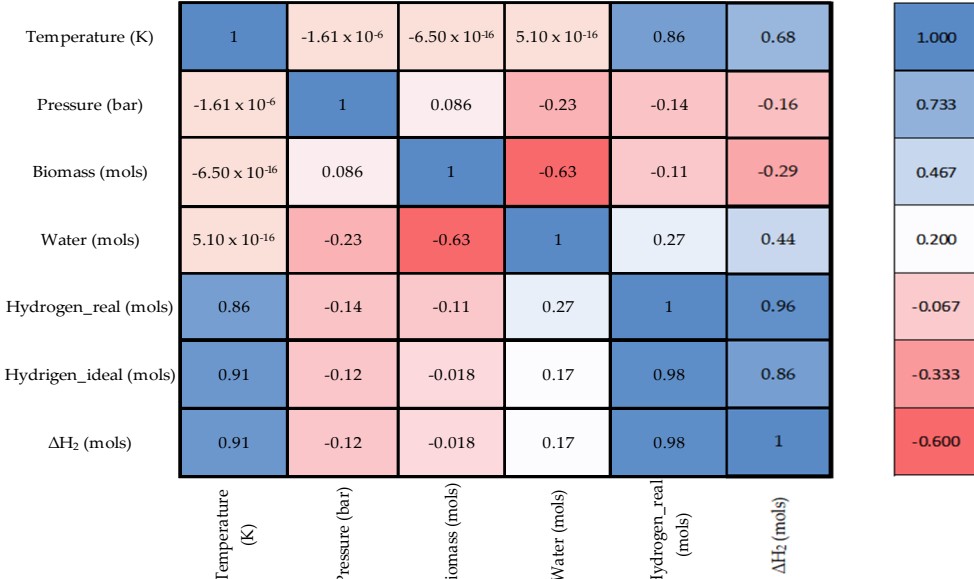

**Figure 9.** Correlation matrix for the final data frame.

### 3.2. Process Monitoring with the Hybrid Model

Simple linear regression was applied, taking as its objective the actual production of hydrogen throughout the process (Hydrogen_real). The simple linear regression took the variables of temperature, pressure, and composition of the biomass/water feed stream as predictor variables. Figure 10 presents the result of the simple linear regression application.

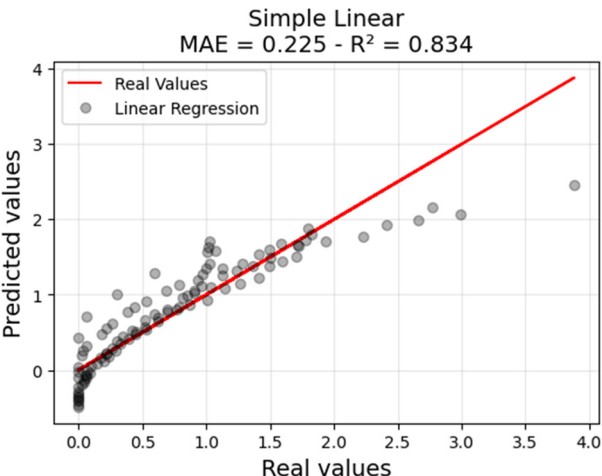

**Figure 10.** Simple linear regression of hydrogen production during the biomass gasification process with supercritical water.

The result indicates that the simple linear regression does not fit the problem in question adequately, considering that it is a non-linear phenomenon.

The next step will be to apply the hybrid modeling methodology, summing the deviation prediction and the value predicted by the ideal model (Hydrogen_ideal). Figure 11 presents the results obtained after the hybridization process.

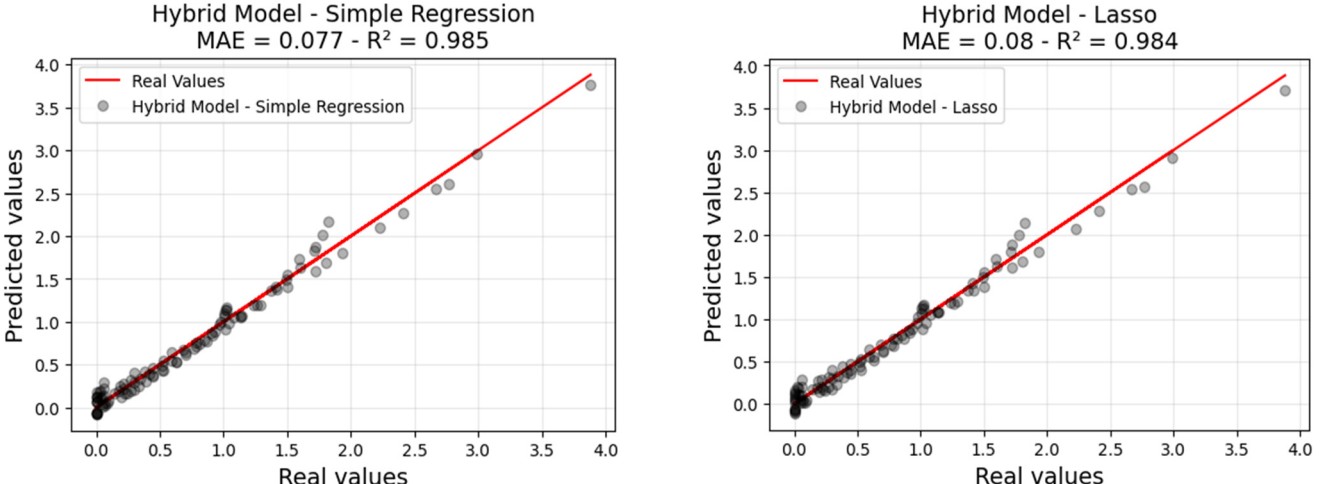

**Figure 11.** Hybrid modeling of biomass gasification process with supercritical water to predict hydrogen production.

The results presented in Figure 11 indicate excellent adjustments with the real data. The hybrid model associating the ideal model with the simple linear regression showed better statistics with a coefficient of determination equal to 0.985 and a mean absolute error equal to 0.07. Table 4 presents a summary of the statistical metrics of the verified models.

**Table 4.** Summary of the statistical metrics of the verified models.

|  | *MAE* | $R^2$ |
|---|---|---|
| Linear Regression | 0.225 | 0.834 |
| Hybrid Model—LASSO | 0.080 | 0.984 |
| Hybrid Model—Linear Regression | 0.077 | 0.985 |

Figure 12 presents a comparison between real data, simulated data considering the reaction system as ideal, and the results obtained from the hybrid modeling, with the simple linear regression model fixing 1 mole of biomass with 5 moles of water in the feed for pressures of 300 and 500 bar.

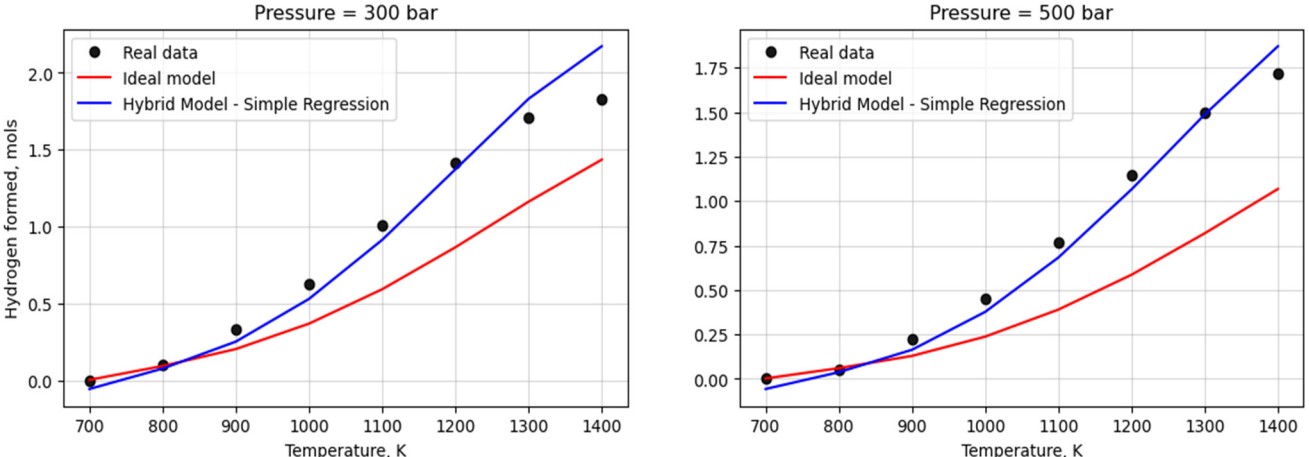

**Figure 12.** Comparison between real data, simulated data considering the reaction system as ideal, and results obtained from the hybrid modeling, with the simple linear regression model fixing 1 mole of biomass with 5 moles of water in the feed for pressures of 300 and 500 bar.

As can be seen in Figure 12, the application of the hybrid modeling proposal considerably improves the ideally simulated data. The ideal model has limitations that make it impossible to predict well the behavior of the system at high pressures, which is verified in Figure 12, as a greater distance between real and calculated data is perceived when the pressure increases from 300 to 500 bar. For both verified pressures, the proposed hybrid model presents excellent results, with coefficients of determination equal to 0.968 and 0.984 for pressures of 300 and 500 bar, respectively.

### 3.3. Conclusions about the Approach and Gains from the Point of View of Process Engineering

The problem used as an example throughout this text deals with a complex reaction with strong non-ideality due to its high temperature and pressure needs, which disfavors the application of simple models such as the ideal gas model. As seen in Figure 7, the ideal model does not present good adjustments concerning the data set used and the deviations tend to be greater with increasing pressure. However, the application of the hybrid model associating the simple linear regression model with the ideal gas model presented good adjustments for the formation of hydrogen under the minimum (300 bar) and maximum (500 bar) pressure conditions verified in this study, thus demonstrating the robustness of this methodology.

Considering that monitoring the formation of hydrogen considering the system as an ideal can be written in a few lines of code, the application of the proposed hybrid modeling described throughout this text has the potential to be applied as an online monitoring tool.

Another advantage consists in the abstraction of non-idealities knowledge. It is not rare that process engineering systems have complex relations, and phenomena that are hard to model, using only a rigorous first-principle-based approach, without incurring the elevated cost of parameter estimation. The hybridization methodology allows the abstraction of these difficulties in the modeling process without losing predictive power.

This work fulfilled the objective of presenting the hybrid modeling architecture as a tool for application in the prediction of industrial processes where a phenomenological model is known that describes the process of interest. The main gain resides in the fact that a data-oriented model can help to correct the deviations caused by the non-ideality of the real phenomena, allowing the use of simplified equations.

## 4. Conclusions

This work proposed and developed a hybridization methodology of engineering models together with data-based models as an alternative to building tools for monitoring and forecasting industrial phenomena. Depending on process complexity, a rigorous approach may be too expensive due to the difficulty in finding adequate parameters that generalize the behavior observed in the plant, or due to the uncertainty associated with the estimates of these parameters.

The case study used as a basis for the development of the methodology was the biomass gasification process using supercritical water as the reaction medium. The proposal is to use linear models, which are simpler and more interpretable, in order to correct the errors committed by an idealized phenomenological model.

Using experimental data, a complex model based on the minimization of Gibbs energy using the cubic Peng–Robinson equation was applied, which presented an excellent fit with the real data, with a lower mean relative deviation of 1.0%. The adjusted phenomenological model was used to augment the database by calculating the equilibrium compositions for different conditions of temperature, pressure, and biomass/water composition in the process feed.

Hydrogen production was adopted as the objective variable, and the next step was the attempt to adjust this variable with a simplified model. The consideration was that the reaction system would behave as ideal; thus, the ideal model was used to adjust the verified process. It presented low adjustment with the real data, presenting values for the mean absolute error equal to 0.281 and 0.322.

Since the ideal model did not fit the actual hydrogen production data well, the application of the hybrid modeling proposal was attempted, using a linear machine learning model to guide the simplified model considered in the prediction of the variable of interest. From this point on, the variable of interest became the error between the calculated results of the ideal values and actual values for hydrogen production.

Two linear regression models were tested for predicting the deviations of the ideal model: simple linear regression and LASSO linear regression. The simple linear regression model showed a better fit when associated with the ideal model for calculating hydrogen production. The predicted deviation values estimated by the data model were added to the results presented for the prediction of ideal hydrogen, and the result of this sum presented good adjustments with the real data. For pressures of 300 and 500 bar, the proposed hybrid model presents excellent results, with determination coefficients equal to 0.968 and 0.984, respectively, thus optimizing the ideal simplified approach.

For comparison purposes, a simple linear regression was applied directly to the variable of interest, the formation of hydrogen. The model presented results for the coefficient of determination equal to 0.834 and an absolute mean deviation equal to 0.225, making the visualization of prediction gains clearer with the application of the hybrid model.

The possibility of using simplified models such as the Clapeyron equation, which are easier to interpret and implement, is a considerable gain, as complex phenomenological models usually demand significant experimental work to determine parameters and have limited generalization, as their reliability is only guaranteed within the limits of experimental conditions.

It was possible to demonstrate how data-based approaches and artificial intelligence can help to improve and give more efficiency to the field of process engineering, allowing the construction of better tools for process monitoring and predictive approaches.

The major challenge found in the process industry is having sufficient quality data to train machine learning approaches. In this work, this obstacle was surpassed through data augmentation through a rigorous equation-of-state (Peng–Robinson) model. However, the lack of a satisfactory amount of data is not a rare situation in the process industry.

In addition, the quality of the data available presents an additional challenge. Industrial data often contain the effects of multiple phenomena, noise, and measurement

uncertainty. This may turn the modeling more difficult because it increases the knowledge incompleteness of the studied processes.

Finally, all the objectives of the work are considered fulfilled, even knowing that there is still much to be done and researched to implement the proposed tools and observe the expected gains.

*Future Work*

The field of industrial digitization is a field of increasing exploration and research, with many opportunities for chemical and process engineers to take a more data-driven view and strengthen the evidence base of arguments.

Possible future work related to this work includes the application of the methodology in real streaming process data and its adaptation to self-learning applications. This could leverage the value generation from industrial data analytics.

This work opens opportunities to explore hybrid methodologies for the use and construction of digital tools for the industry. Opportunities are focused on exploring how the model behaves against real data on the conversion of biomass into hydrogen during the process of supercritical biomass gasification.

**Author Contributions:** J.M.d.S.J., project proposal; Í.A.M.Z., methodology development; J.M.d.S.J. and Í.A.M.Z., research and validation; J.M.d.S.J. and Í.A.M.Z., development of results; J.M.d.S.J. and Í.A.M.Z., constant evaluation of results; A.P.M., supervision and guidance throughout the development of the article. All authors have read and agreed to the published version of the manuscript.

**Funding:** This research did not receive external funding.

**Institutional Review Board Statement:** Not applicable.

**Informed Consent Statement:** Not applicable.

**Data Availability Statement:** The data used in this work were obtained from simulations based on the thermodynamic approach as described. Similar results can be obtained in any process simulator and the treatment from a machine learning point of view can be easily replicated. We encourage everyone to use the architecture described in any possible problem where you have knowledge of data from any process (real or rigorously simulated) and data obtained from simplified modeling. The purpose of the text is not the verified system but the hybrid approach that allows associating machine learning models with phenomenological models for monitoring processes.

**Acknowledgments:** The authors would like to thank the entire faculty of the State University of Campinas for their contribution to the personal and professional development of countless lives and all the professors who support the development of society. In addition, the authors thank Radix Engineering and Software for providing the necessary time and tools demanded by the development of this methodology.

**Conflicts of Interest:** The authors declare no conflict of interest.

## Nomenclatures

| | |
|---|---|
| G | Total Gibbs energy |
| l | Liquid phase |
| s | Solid phase |
| v | Vapor phase |
| NC | Number of components |
| NF | Number of phases |
| $n_i^k$ | Number of moles of component i in phase k; i = [1, 2, 3, . . . , NC]; k = [v, l, s] |
| R | Universal gas constant |
| T | Temperature |
| P | Pressure |
| $\mu_i^k$ | Chemical potential of component i in phase k; i = [1, 2, 3, . . . , NC]; k = [v, l, s] |
| $\hat{f}_i^k$ | Fugacity of component i in phase k |

| | |
|---|---|
| $f_i^o$ | Fugacity of pure species i in a standard reference state |
| $a_{mi}$ | Number of atoms of element i in component m |
| $n_i^o$ | Number of moles in standard state |
| $H_i^k$ | Enthalpy of component i in phase k |
| $H_i^0$ | Enthalpy of component i in the standard state |
| $H^0$ | Total enthalpy |
| $Cp_i^k$ | Heat capacity of component i in phase k; i = [1, 2, 3, . . . , NC]; k = [v, l, s] |
| $\mu_i^0$ | Chemical potential of component i in a standard reference state |
| $\hat{\varnothing}_i^k$ | Coefficient of fugacity of component i in phase k; i = [1, 2, 3, . . . , NC]; k = [v, l] |
| $y_i$ | Mole fraction of component i in the vapor phase |
| $x_i$ | Molar fraction of component i in the liquid phase |
| $P_i^{sat}$ | Component saturation pressure i |
| $a_i, b_i, c_i$ | Constants for calculating component saturation pressure i |
| $A_{n,i}$ | Constants for calculating the heat capacity of the component i in the vapor phase. i = [1, 2, 3, . . . , NC]; k = [1, 2, 3 and 4] |
| $A_i, B_i, C_i, D_i$ | Constants for calculating the heat capacity of component i in the solid phase |
| $Z_i$ | Compressibility factor |
| A, B, u, w | Parameters of the cubic equation of state |
| $a_m$ | Attraction parameter for mixtures |
| $b_m$ | Repulsion parameter for mixtures |
| $k_{ij}$ | Binary interaction parameter |
| $T_{c,i}$ | Critical component temperature i |
| $P_{c,i}$ | Critical component pressure i |
| $w_i$ | Acentric factor |
| M | Constant for Kamath, Biegler, and Grossmann constraints |
| $\sigma^k$ | Slack variables for Kamath, Biegler, and Grossmann constraints |
| n | Number of moles |
| $H_2^k$ | Moles of hydrogen; k = [real, ideal, prredict] |
| $\dot{y}$ | Actual value of the target variable |
| $\hat{y}$ | Estimated value of the target variable |
| $\overline{y}$ | Average value of the target variable |

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
