# Peer review of "Hybrid Modeling of Machine Learning and Phenomenological Model for Predicting the Biomass Gasification Process in Supercritical Water for Hydrogen Production"

_2673-4117, doi:10.3390/eng4020086_

Round 1

Reviewer 1 Report

The manuscript entitled “Hybrid Modeling of Machine Learning and Phenomenological 2 Model for Predicting the Biomass Gasification Process in Super-3 Critical Water for Hydrogen Production” Presents quite a novel concept, but still, the overall structure of the paper is not good enough and as per the first impression the paper needs a profound and thorough revision of the manuscript linked with the revised structuring. Hence, overall the manuscript needs a major revision before being accepted for publication. Following are some of the pointers

1.      The abstract seems to not give a clear point of view of what the author has achieved in this study. No results have been given in the abstract. A thorough modification of the abstract is required strictly.

2.      Line 40 “ Biomass gasification processes in supercytic water……” what is the author wanted to state in this line? Kindly please explain the point of view

3.      Line 43-46 requires modification and reference

4.      Line 56-57 add reference

5.      In the section 1.1. The process of gasification of biomass in supercritical water the first three paragraphs are required to be revisited it seems to make the audience confuse as what the author wanted to state under the heading

6.      In line 74-75 the author stated “there is no kinetic model robust that accurately describes all 74 the details of the reaction mechanism…..” however in the corresponding line the author have stated the kinetic model study from the literature. Can you please elaborate your point of view.

7.      The introduction section is lacking structuring in a proper way which includes (i) problem statement, (ii) proposed technological point, (III) selection of this technology based on literature review (and author have just stated a single reference to explain his point of view), (IV) the most important is the objective and novelty explanation of this study missing.

8.      Line 90 “phenomenological  point of view….” What do you mean by phenomenological point ?? can you explain it a bit more so it may be easy for the audience to understand.

9.      Line 96 “this methodology…..” which methodology is author referring to in this statement

10.   It is explicitly important to state initially under the methodology heading what shall be the main route the author would be taking to explain his point of view in a concise way. Rather no point can be found in this regarding the approach adopted in this study.

11.   It would be better if the machine learning approach adopted in the optimization process should be explained or the framework adopted for the study

12.   It is still unclear in section 2.3 how author manage to link experimental data using a phenomenological model?? Its quite strange but no information can be found which process model platform author have utilized for their study that establishes a link between the Gibbs energy minimization with the equation of state Peng Robinson model.

13.   Section 2 is lacking with the references please add few more recent references to support your point of view

14.   In section 3.3 “Conclusions about the approach and gains from the point of view of process engineering” but no information related to the process engineering aspect have been depicted in the results rather a detailed numerical modelling approach have been shown. So the author need to defend his point of view in a more clear way as it seems it not related with process engineering aspect.

15.   If the author can enlist in the conclusion section what are the major barrier in the pathway of the current study that consider as a barrier to this study.

16.   It would be better future scope can be separated in a separate heading 

Need improvement. 

Author Response

The authors are grateful for the attention dedicated to our article. All notes were valuable and all were attended to.

The letter below sets out the discussions in more detail.

Reviewer 2 Report

The authors present a machine learning-based hybrid modeling for predicting the biomass gasification process in super- 3 critical water for hydrogen production. Overall, the manuscript is well structured, the relevant background and the need of the approach to circumvent the challenges are clearly stated. I would like to recommend it for publication after the authors consider the following revision:

1.     In recent studies, machine-based hybrid modeling approaches have been applied to various systems (e.g., Energy, 2023, 273, 127126). Therefore, it would be beneficial to provide a brief overview of these studies in the introduction.

2.     It would be helpful if the author could provide further details regarding the mathematical formulation and solution of the equilibrium problem, such as the variables, constraints, and solution time of the model in GAMS, and explain why the CONOPT solver was chosen over other solvers like LINDOGLOBAL and CPLEX.

3.     Equations (17-20) require further clarification, including an explanation of the meaning of the variables and parameters used in these equations.

4.     Is it possible to integrate this hybrid model in process simulators such as Aspen simulation tools?

5.     To ensure the reproducibility of the results, it is highly recommended that the authors provide detailed calculation and modeling data, such as in the supporting material.

6.     The manuscript contains minor errors that should be corrected, such as 22,032% (line 204) and Equation (20). The authors should thoroughly check the manuscript for these types of errors.

No

Author Response

(The authors gave the same response as above.)

Round 2

Reviewer 1 Report

1. Please underscore the scientific value-added to your paper in your abstract. Your abstract should clearly state the essence of the problem you are addressing, what you did and what you found and recommend. That will help a prospective reader of the abstract to decide if they wish to read the entire article. 2. In your discussion section, please link your empirical results with a broader and deeper literature review. 3. In addition, the below reference can be further incorporated to improve the literature review in the paper.
  1. https://doi.org/10.1016/j.fuel.2023.128458
  2. https://doi.org/10.1016/j.resconrec.2022.106847 
  3. https://doi.org/10.1016/j.fuel.2023.128635
  4. https://doi.org/10.3389/fenrg.2021.782139
  5. https://doi.org/10.1016/j.fuel.2022.125718
  6. https://doi.org/10.1016/j.renene.2018.07.142

If possible, it can be improved further. 

Author Response

The authors are grateful for the reviewer's precious contribution to this text. The notes were read and attended to.
